# Domain-Informed Negative Sampling Strategies for Dynamic Graph Embedding in Meme Stock-Related Social Networks

## Abstract

Social network platforms like Reddit are increasingly impacting real-world economics. Meme stocks are a recent phenomena where price movements are driven by retail investors organising themselves via social networks. To study the impact of social networks on meme stocks, the first step is to analyse these networks. Going forward, predicting meme stocks' returns would require to predict dynamic interactions first. This is different from conventional link prediction, frequently applied in e.g. recommendation systems. For this task, it is essential to predict more complex interaction dynamics, such as the exact timing and interaction types like loops. These are crucial for linking the network to meme stock price movements. Dynamic graph embedding (DGE) has recently emerged as a promising approach for modeling dynamic graph-structured data. However, current negative sampling strategies, an important component of DGE, are designed for conventional dynamic link prediction and do not capture the specific patterns present in meme stock-related social networks. This limits the training and evaluation of DGE models in analysing such social networks. To overcome this drawback, we propose novel negative sampling strategies based on the analysis of real meme stock-related social networks and financial knowledge. Our experiments show that the proposed negative sampling strategy can better evaluate and train DGE models targeted at meme stock-related social networks compared to existing baselines.

## CCS Concepts

• **Networks** → **Network dynamics**; • **Information systems** → **Social networks**.

## Keywords

Dynamic Graph Embedding; Negative Sampling Strategies; Social Network Analysis; Reddit; Wallstreetbets

**ACM Reference Format:**
Anonymous Author(s). 2025. Domain-Informed Negative Sampling Strategies for Dynamic Graph Embedding in Meme Stock-Related Social Networks. In *Proceedings of ACM Web Conference 2025 (WWW '25)*. ACM, New York, NY, USA, 12 pages. https://doi.org/XXXXXXX.XXXXXXX

## 1 Introduction

Social networks play an ever increasing role in society [1, 28]. Various studies show that social networks, such as Twitter [14] and Reddit [33], also influence financial markets. The research presented in this paper is motivated by Reddit and the GameStop (GME) market frenzy occurring around January 2021 when users on the subreddit 'Wallstreetbets' discussed GME and collectively caused a market frenzy [31]. It has become clear that internet users are a notable group influencing stock prices specifically for so called 'meme stocks' [8], stocks that receive significant attention on social media. To study the relationship between retail investors on social networks such as Reddit and the stock markets, a thorough understanding is needed of the network structure and behavior of the people posting on these networks. A first step to studying the relationship is in understanding changes in posting behavior over time which may trigger stock market action [31]. For this purpose, a dynamic network model is needed which captures the dynamics of posting behavior at the individual node level. Such models need to be scalable due to the large number of users and high volume of interactions and posts in social networks.

Dynamic graph embedding (DGE) has emerged as an effective tool for tacking these challenges [37, 40]. Graphs naturally describe social networks by representing individuals as nodes and their interactions as edges, providing a structured framework for analysis. DGE builds on this by transforming the nodes and edges into continuous vector representations (node embeddings), preserving both the network's structural and temporal properties. This approach allows DGE to capture the dynamic evolution of social networks over time, enabling insights into complex user interactions and facilitating predictions of network behavior.

For DGE, dynamic link prediction (DLP) is an important component which predicts if there is a link between two nodes based on their embeddings [2]. From a technical perspective, DLP can evaluate the quality of generated embeddings and serve as the training objective. From an application perspective, predicting when two users will interact in the future based on embeddings can help identify stock market trends [5, 19, 44], such as renewed interest in a stock, which may manifest in a new stock market frenzy.

DGE models need to accurately predict both existing interactions (positive samples) and nonexistent connections (negative samples) [9, 20]. If the model only predicts that all interactions exist, it may achieve good performance on positive samples but will incorrectly identify nonexistent connections, leading to unreliable and misleading results. However, in meme stock-related social networks, the number of negative samples far exceeds that of positive samples. Due to the huge size of these social networks, users typically only communicate with a fraction of other users [13, 35]. Therefore, the majority of these negative samples provide little valuable information, as many users may never interact. We conjecture that using such obvious non-connections for model training

and evaluation will focus the models prediction ability upon these obvious non-connections, whereas the real challenge lies in predicting negative samples which are difficult to predict in real social networks. This highlights the need to carefully select informative negative samples, a process known as negative sampling [41].

Most existing negative sampling strategies for DLP are primarily based on random or heuristic approaches [6, 26, 27]. For instance, random negative sampling is one of the most widely used strategies [26]. It generates one negative sample for each positive sample $(u, v, t)$, where $u, v, t$ is the sender, receiver, and occurrence time of the interaction, by replacing $v$ with a random user. Such strategy leads to many of the obvious non-connections to be part of the generated negative samples which results in deceivingly outstanding performance. State-of-the-art (SOTA) dynamic graph embedding (DGE) models can achieve the AUC (Area Under the Receiver Operating Characteristic Curve) over 0.9 on certain datasets when trained and evaluated using this negative sampling strategy [11, 29, 45]. However, the practical use for real applications, such as meme stock-related social network prediction, is low.

For illustration, consider the use case of predicting when the users who have already interacted (i.e., there are edges connecting two nodes) will interact again. This is important because repeated interactions often indicate renewed interest or users' joint and repeated interest in a stock, which can lead to price movements for meme stocks [31]. To evaluate the model's prediction ability in such a case, we generated three types of negative samples for each positive sample $(u, v, t)$: $(u, v, t + 6h)$, $(u, v, t + 12h)$, and $(u, v, t + 24h)$. These samples test whether the model can correctly predict if nodes that have interacted will interact again after 6, 12 and 24 hours. We use the dataset AMC (see Section 4) and the SOTA DGE model Temporal Graph Networks (TGNs) [29] as an example. The results in Table 1 show that the TGNs achieved an AUC of 0.9736 when trained and tested using random negative sampling, closely matching results reported in the original paper [29]. However, the performance dropped strongly when tested with the other three types of negative samples. This indicates that random negative sampling strategy limits the DGE model's ability to accurately predict when previously interacting nodes will interact again.

**Table 1: Test AUC of TGN with various negative sampling strategies (Using AMC Dataset, January for Training and February for Validation and Testing)**

| Strategy | Random | 6h | 12h | 24h |
|---|---|---|---|---|
| AUC | 0.9736 | 0.6041 | 0.6982 | 0.7681 |

With this example, we show that the design of the negative sampling strategy should be closely tied to domain knowledge. This has also been proved by a recent study study [23]. In the settings like meme stock-related social networks, interactions between users are not random or uniform. A generic negative sampling strategy may miss some important information, leading to suboptimal performance in predictive tasks. By incorporating domain-specific knowledge, such as understanding the significance of predicting the exact time of repeated interactions, a more effective negative sampling strategy can be developed.

In this paper, we analyzed three real-word meme stock-related social networked datasets containing interactions on Reddit related to three companies, GameStop (GME), American Multi-Cinema (AMC), and BlackBerry (BB), and identified several key characteristics of meme stock-related social networks, such as the frequency of interactions between users, and the presence of unique interaction types such as loops. Based on these insights, we developed several individual negative sampling strategies specifically tailored to these network properties. Each strategy captures a distinct aspect of the network dynamics. We also developed a joint negative sampling strategy, incorporating these individual negative sampling strategies. To overcome the complexity and imbalance between positive and negative samples caused by incorporating all of these negative samples into the training process, we implemented positive enhancement where additional positive samples are included during training to maintain a balanced ratio between positive and negative interactions.

In summary, with this paper we make the following contributions:

(1) We explored the application of DGE models to a special type of special social networks, meme stock-related social networks. We found that the current design of negative sampling strategies, an important component of dynamic graph embedding models, limits the performance of DGE models in this kind of social networks.

(2) We proposed several individual negative sampling strategies based on the analysis of three real-word meme stock-related social networks and corresponding financial domain knowledge. Each of them evaluates a certain part of DGE models' prediction ability in meme stock-related social networks. We also proposed a negative sampling strategy named **D**(omain)**I**(nformed)**N**(egative)**S**(ampling) that combines these single strategies and further balances the positive and negative sample by positive enhancement.

(3) We conducted extensive experiments to show the effect of negative sampling strategies in the evaluation and training of DGE models. The experimental results also show that our proposed negative sampling strategies can improve DGE model's prediction performance in meme stock-related social networks.

## 2 Preliminaries

In this section, we define the representation of dynamic social networks and negative sampling strategy formally. In addition, we briefly introduce dynamic graph embedding and link prediction.

### 2.1 Dynamic Social Network Representation

As we introduced in Section 1, graphs can be used to represent social networks. In DGE, dynamic graphs can be either continuous or discrete. Continuous dynamic graphs provide higher time resolution which allows for a more accurate representation of the temporal evolution of networks [18, 40]. Hence, we represent a dynamic social network with a continuous dynamic graph.

DEFINITION 1 (CONTINUOUS DYNAMIC GRAPH). *A dynamic graph is denoted by $\mathcal{G} = (\mathcal{V}, \mathcal{E})$, where $\mathcal{V}$ is the node set containing $n$ nodes and $\mathcal{E} = \{(u_i, v_i, t_i) \mid 1 \leq i \leq m\}$, where $0 \leq t_1 \leq t_2 \leq \cdots \leq t_m = T$, is the edge set containing $m$ directed edges. The source node, destination node, and the timestamp of an edge are denoted by*

$u_i \in \mathcal{V}$, $v_i \in \mathcal{V}$, and $t_i$ respectively. $T$ is the latest timestamp of the observed period.

## 2.2 Dynamic Graph Embedding and Link Prediction

DEFINITION 2 (DYNAMIC GRAPH EMBEDDING). *A dynamic graph embedding model is a function that maps a dynamic graph $\mathcal{G} = (\mathcal{V}, \mathcal{E})$ to a time-dependent continuous vector space. It assigns each node $u \in \mathcal{V}$ a time-specific embedding $z_u(t) \in \mathbb{R}^d$, where $d$ is the dimension of the embedding. The node embeddings should preserve the evolving relationships and interactions between nodes over time.*

DEFINITION 3 (DYNAMIC LINK PREDICTION). *Given a dynamic graph $\mathcal{G} = (\mathcal{V}, \mathcal{E})$, dynamic link prediction aims to predict whether a future edge $(u, v, t)$ $(u \in \mathcal{V}, v \in \mathcal{V}$ and $t > T)$ exits based on $\mathcal{G}$.*

DLP can be based on embeddings because the DGE model captures the evolving relationships between nodes in a continuous vector space. These embeddings represent how nodes interact over time, allowing DLP to predict future connections by analyzing the similarity or changes in the node embeddings.

## 2.3 Negative Sampling Strategy in DLP

DEFINITION 4 (NEGATIVE SAMPLING STRATEGY). *Given a continuous dynamic graph $\mathcal{G} = (\mathcal{V}, \mathcal{E})$, the set of all possible negative samples is defined as: $\mathcal{E}_{neg} = \{(u_i, v_i, t_i) \mid u_i, v_i \in \mathcal{V}, (u_i, v_i, t_i) \notin \mathcal{E}, 0 \le t_i \le T\}$. A negative sampling strategy is a method for selecting a subset $\mathcal{E}'_{neg} \subseteq \mathcal{E}_{neg}$ to use during training or evaluation.*

## 3 Related Work

In this section, we review the dynamic graph embedding methods and present an analysis of existing negative sampling strategies.

## 3.1 Dynamic Graph Embedding Models

According to the survey of Barros et al. [2], learning-based DGE models have become the dominant approach in the field today. Thus, we focus on these learning-based models in this paper.

Learning-based DGE models leverage deep learning techniques, such as recurrent neural networks (RNNs) [10], graph neural networks (GNNs) [30], and attention mechanisms [34], to capture the evolving relationships in dynamic networks. One prominent category of these models incorporates memory mechanisms which store and update node-specific information over time to better capture temporal dependencies [24, 29, 32, 36]. In addition to memory-based models, other approaches leverage advanced architectures like transformers, such as DyGFormer [42] and GraphERT [3]. Cong et al. claim that complex neural networks such as RNNs and attention mechanism are not always necessary, and proposed Graph-Mixer that relies on multi-layer perceptrons (MLPs) [7]. For these DGE models, the majority [3, 24, 29, 32, 36, 42] uses DLP as one of the tasks to evaluate the quality of generated embeddings. Most models [7, 29, 36, 42] use DLP as the learning objective.

## 3.2 Negative Sampling Strategy in DLP

The most common negative sampling strategy used is Random Negative Sampling [26]. To generate negative samples, the destination node $v$ of each positive sample $(u, v, t)$ is replaced with a random node selected from all nodes. Random Negative Sampling is employed by the majority of studies developing DGE models. Though it is a straight-forward method to implement the generated negative samples are mostly uninformative because the two nodes are likely to have never interacted before and therefore have completely different features.

Recent works suggest that better negative sampling strategies are needed for DLP. For example, Poursafaei et al. [26] argue that two nodes may connect multiple times. To address this, they proposed Historical Negative Sampling. They generate negative samples $(u', v', t')$ by requiring that node $u'$ and $v'$ have been connected at some time before $t'$. These negative samples do provide more information compared to random sampling, but they still focus on one specific aspect of the network.

Some studies [6, 12] use the idea of curriculum learning. They first generate all negative samples. Then, they select more difficult negative samples as model training progresses according to a criteria they defined for measuring the difficulty of negative samples. When generating negative samples, they replace $v$ of each positive sample $(u, v, t)$ with all nodes except nodes $u$ and $v$. If a node never or rarely becomes a source node, its relationship with other nodes is then not well captured by these negative samples. When designing the difficulty measurement criteria, they lack the consideration of domain knowledge. The negative sampling strategy proposed by Poursafaei et al. [27] also considers all negative samples and reduces the number of negative samples by merging some samples happens during closing time. However they do not consider the situation that some nodes never or rarely become a source node.

Overall, the existing strategies show the following deficiencies: 1)Sample large of node pairs that never interacted, providing little useful information, or only focus on previously connected nodes, ignoring relationships between nodes that have never interacted; 2) Ignore the fact that nodes in social networks can act as both sender and receiver and that these roles often inter change; 3) Most approaches fail to incorporate domain knowledge, which could optimize the sampling process for specific network characteristics. Thus, these negative samples cannot well evaluate or train DGE models used for predicting meme stock related social networks.

To address these deficiencies, we analyse three real-world meme stock-related social networks in combination with financial domain knowledge, to design negative sampling strategies that can better capture these networks.

## 4 Datasets

We study three meme stock-related social network datasets [43] collected from WallStreetBets (WSB), shown as r/wallstreetbets on Reddit, which is a financial community where participants discuss investments. These three datasets include interactions regarding three companies: GameStop (GME), American Multi-Cinema (AMC), and BlackBerry (BB). The stock prices of all three companies were strongly influenced by these interactions during the time when these interactions happened [4, 25, 31].

Reddit uses a post-comment structure, where user behavior falls into two categories. First, a user can create a post. Second, a user can comment on an existing post. Therefore, in the original datasets, users are treated as nodes ($\mathcal{V}$) and their interactions form the

directed edges ($\mathcal{E}$). When a user creates a post, this is represented as a loop, forming an edge from the user to themselves ($u, u, t_i$). When user A comments on a post made by user B, this creates a directed edge from node $u$ to node $v$, denoted as ($u, v, t_i$), where $t_i$ is the timestamp of the interaction in UTC format.

We adjusted the original dataset as follows: 1) Removed unknown users and excluded data from months with excessively sparse interactions; 2) Reduced the time resolution of these datasets to 5 minutes. This means that all timestamps within each 5-minute interval were grouped and assigned the same timestamp. The descriptive statistics of the processed datasets are presented in Table 2.

**Table 2: Descriptive statistics of three datasets. Unique node pairs refer to interactions between two distinct users, where the direction of the interaction matters. Loops represent edges whose source and destination node are the same.**

| Dataset | Nodes $|\mathcal{V}|$ | Edges $|\mathcal{E}|$ | Unique Node Pairs | Loops | Strat Date | End Date |
|---|---|---|---|---|---|---|
| GME | 517,975 | 3,976,267 | 2,692,485 (67.71%) | 134,010 (3.37%) | 2020-09-01 | 2021-08-31 |
| AMC | 313,006 | 2,207,981 | 1,544,006 (69.92%) | 192,917 (8.73%) | 2021-01-01 | 2021-12-31 |
| BB | 104,453 | 406,916 | 305,349 (75.03%) | 30,434 (4.47%) | 2021-01-01 | 2021-12-31 |

## 5 Methodology

In this section, we first propose three individual negative sampling strategies. Each strategy captures a particular aspect of the network and is designed based on specific characteristics of meme stock-related social networks as well as financial domain knowledge. Then we proposed a joint negative sampling strategy that combines these individual strategies and balances the positive and negative samples. These strategies are visualized in Figure 1.

In practical applications, learning-based DGE typically divides the edge set $\mathcal{E}$ into multiple batches, containing a fixed number of edges, for processing. As shown in Figure 1, all interactions within a batch are processed at once, with each batch being handled sequentially. This allows for more efficient computation and better memory management, especially when dealing with large datasets. Consequently, negative sampling is also performed on a per-batch basis. Thus, for illustration, we consider negative sampling on a batch $\mathcal{E}_{batch} = \{(u_i, v_i, t_i) \mid b \le i < b+k\}$ containing $k$ interactions starting from the $b$-th interaction. We additionally define $\mathcal{ET}_{batch} = \{t_i \mid b \le i < b+k\}$ containing all timestamps in $\mathcal{E}_{batch}$.

### 5.1 Individual Negative Sampling Strategies

*5.1.1 Random sender and receiver.* We begin with the most fundamental evaluation, predicting the relationship between two nodes. In this negative sampling strategy, we do not strictly test the model's ability to predict when two nodes will interact, but to evaluate whether the model can accurately predict if an interaction will happen between two nodes.

The random negative sampling strategy [26], introduced in Section 3, also serves this purpose and works in bipartite datasets, where source nodes represent users and destination nodes represent items, such as products or services. While effective for predicting

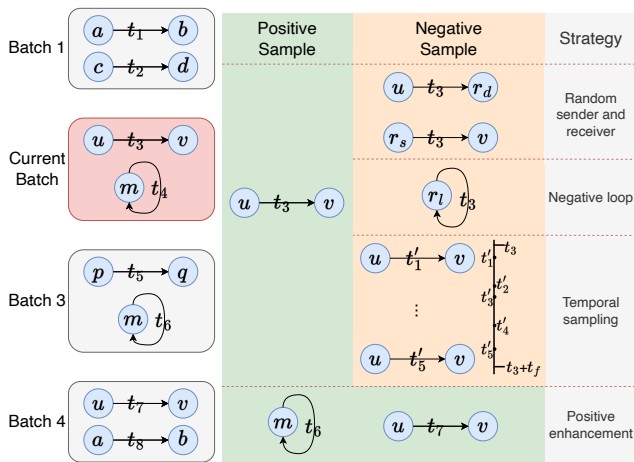

**Figure 1: Visualization of proposed negative sampling strategies. Taking a dynamic network with 8 interactions as an example. The batch size is 2, meaning that the interactions are processed in groups of two. The showed negative samples are generated according to positive sample ($u, v, t_3$).**

the next item a user might be interested in, this method falls short in the context of meme stock-related social networks, where the relationships between any two nodes are of interest [15]. In such networks, a node can act as both a source node and a destination node, and these roles often interchange frequently.

To solve this problem, we generate negative samples by randomly replacing both the source and destination nodes of each positive sample. Specifically, for each positive sample $(u, v, t) \in \mathcal{E}_{batch}$, two negative samples are generated: $(u, r_d, t)$ and $(r_s, v, t)$ where both $r_d$ and $r_s$ are randomly selected from $\mathcal{V} \setminus \{u, v\}$.

This strategy addresses the shortcomings of random negative sampling by enhancing the model's ability to evaluate the potential for interactions between any two nodes in a network, regardless of their typical roles as source or destination. By this the model predicts the next likely interaction based on meme stock-related social network dynamics. If these interactions occur in bulk, it may spill over into meme stock market activity [31].

*5.1.2 Temporal sampling.* Temporal sampling additionally tests the model's ability to predict the exact time when two nodes will interact. As shown in Table 2, the unique node pairs constitute only 67.71%, 69.96% and 75.03% of the total number of edges in GME, AMC and BB, respectively. This shows that a large number of node pairs interacted more than once, while others interact only once. In meme stock-related social networks, accurately predicting when users who have already interacted will interact again is crucial. This is because repeated interactions often signal renewed interest or joint and sustained interest in a stock, which can influence price movements in meme stocks.

Temporal sampling works by generating negative samples for node pairs that have interacted in the past, but at future time points where no interaction has occurred. Specifically, for each positive sample $(u, v, t) \in \mathcal{E}_{batch}$, $q$ negative samples are generated: $\{(u, v, t_n) \mid 1 \le n \le q\}$ where $(u, v, t_n) \notin \mathcal{E}$ for any $n$, and the

timestamps $t_n$ are uniformly and randomly distributed within the interval $[t, t + t_f]$. To adapt to different data sets, $q$ and $t_f$ are set as adjustable parameters. However, to avoid information leakage, $t + t_f$ should not be greater than the largest timestamp in $\mathcal{ET}_{batch}$ i.e., $t + t_f \leq \max(\mathcal{ET}_{batch})$.

Temporal sampling evaluates the model's ability to predict interaction timing by introducing negative samples at future timestamps where no interaction has yet occurred. By focusing on previously interacting node pairs and generating future interactions that have not happened, the strategy tests the model's capability to identify true future interactions among potential ones.

*5.1.3 Negative loops.* As we discussed in Section 4, due to the post-comment structure used by Reddit, loops account for 3.37%, 8.73%, and 4.47% of interactions in GME, AMC and BB dataset, respectively (cf. Table 2). Predicting the existence of a loop is important because it may indicate that the user is re-engaging, potentially driven by new developments or shifts in stock performance.

This negative sampling strategy is designed to evaluate if a DGE model can well predict the existence of loops. Specifically, for each timestamp $t \in \mathcal{ET}_{batch}$, one negative sample is generated: $(r_l, r_l, t) \notin \mathcal{E}$ where $r_l$ is a random node that has not formed a self-loop, i.e., $(r_l, r_l, t')$ does not exist for any $t' < t$. We require $r_l$ to be a node that has not formed a self-loop in order to specifically evaluate the model's ability to predict whether a node that is not expected to form a loop will indeed do so. The evaluation of nodes that are expected to form loops is already covered by the temporal sampling strategy.

## 5.2 Combination

The above introduced individual negative sampling strategies each captures a specific aspect of the dynamic graph. In this subsection, we explain how these strategies can be combined to create a more comprehensive and effective negative sampling strategy.

DLP is not only an evaluation tool but also serves as a crucial task during the training process for many DGE models. During the evaluation phase, all available negative samples can be used to fully assess the model's performance. However, in the training process, using too many negative samples can strongly skew the data distribution and create an imbalance [16, 22]. This leads to the model becoming biased toward predicting negative outcomes, which reduces its ability to correctly identify positive interactions.

To address this issue, we proposed a negative sampling strategy named DINS, designed to combine all these individual negative sampling strategies effectively while maintaining a balanced approach during training, which is detailed in Algorithm 1.

To balance the positive and negative samples, positive enhancement is conducted as follows (line 15-22 in Algorithm 1): for each edge $(u, v, t)$ happening after the current batch, we check if nodes $u$ and $v$ interacted within the current batch. If so, the positive sample $(u, v, t)$ is added to the current sample set. However, the number of added positive samples does not exceed the size of the current batch to avoid increasing the overall training time or introducing unnecessary computational overhead. This prevents the model from becoming biased towards predicting that edges are always nonexistent, ensuring that the model learns to accurately distinguish

between the presence and absence of edges, rather than defaulting to negative predictions.

---

**Algorithm 1** Negative Sampling Strategy for Training

---

1: **Input**: $\mathcal{E}_{batch}$, $\mathcal{V}$, $\mathcal{ET}_{batch}$, $t_f$, $q$, $k$
2: $\mathcal{S} \leftarrow \emptyset$ {Initialize a collection for all samples }
3: **for** $e = (u, v, t)$ **in** $\mathcal{E}_{batch}$ **do**
4: $r_s, r_d \leftarrow$ **random**$(\mathcal{V} \backslash \{u, v\})$, **random**$(\mathcal{V} \backslash \{u, v\})$
5: $\mathcal{S} \leftarrow \mathcal{S} \cup \{(r_s, v, t)\}$ {Random sender}
6: $\mathcal{S} \leftarrow \mathcal{S} \cup \{(u, r_d, t)\}$ {Random receiver}
7: $t_1, \cdots, t_q \leftarrow$ **random**$([t, t + t_f])$
8: $\mathcal{S} \leftarrow \mathcal{S} \cup \{(u, v, t_1), \cdots, (u, v, t_q)\}$ {Temporal sampling}
9: **end for**
10: $\mathcal{V}_l \leftarrow$ nodes have never formed a loop
11: **for** $t$ **in** $\mathcal{ET}_{batch}$ **do**
12: $r_l \leftarrow$ **random**$(\mathcal{V}_l)$
13: $\mathcal{S} \leftarrow \mathcal{S} \cup \{(r_l, r_l, t)\}$ {Negative loops}
14: **end for**
15: $\mathcal{E}_{after} \leftarrow \{(u, v, t) | t > \max(\mathcal{ET}_{batch})\}$
16: positive_count $\leftarrow 0$
17: **for** $e = (u, v, t)$ **in** $\mathcal{E}_{after}$ **do**
18: **if** $\exists t' s.t. (u, v, t') \in \mathcal{E}_{batch}$ **and** positive_count $< k$ **then**
19:  $\mathcal{S} \leftarrow \mathcal{S} \cup \{(u, v, t)\}$ {Positive enhancement}
20:  positive_count $\leftarrow$ positive_count$+1$
21: **end if**
22: **end for**
23: **Output** $\mathcal{S}$

---

## 6 Experiments

In this section, we validate our proposed DINS strategy experimentally. All experiments are conducted on a machine with an Intel Xeon Platinum 8360Y (2.4 GHz, 18 cores), 128 GiB DDR4 RAM, and a NVIDIA A100 (40 GiB HBM2 memory), running Linux release 8.6.

## 6.1 Dynamic Graph Embedding Models

To demonstrate that our negative sampling strategy can generalize and perform well regardless of the specific model architecture, we selected three distinct dynamic graph embedding models with varying designs:

1) **TGNs**[29] is one of the most widely recognized DGE model that utilize memory mechanisms. TGNs has gained wide-spread attention due to its high performance on various datasets, making it a strong representative of memory-based DGE models.

2) **DyGFormer** [42] employs advanced transformer architectures, which allow for more sophisticated modeling of temporal dependencies in dynamic graphs. The authors of DyGFormer claim that it outperforms SOTA DGE models in various tasks, making it a strong candidate for evaluating the effectiveness of our proposed negative sampling strategies.

3) **GraphMixer** [7] achieves comparable or even superior performance using multi-layer perceptrons (MLP) instead of complex architectures used by other models, while also converging more quickly. This makes it an ideal choice for testing our negative sampling strategy across different types of DGE models, allowing us to assess its effectiveness on simpler yet efficient models.

Dynamic graph embedding (DGE) can be either transductive or inductive [39]. Transductive learning limits predictions to nodes in the training set, while inductive learning allows the model to generalize and predict for unseen nodes. As we do not know beforehand whether a new user will be introduced to the network, we adopt the transductive approach in all experiments.

## 6.2 Experimental Setting

*6.2.1 Baselines.* We select two strategies as baselines.

**Random Negative Sampling (Negative)** is widely used by most dynamic graph embedding studies [7, 29, 36, 42]. Choosing this as a baseline allows us to establish a standard for comparison, ensuring that our proposed methods are evaluated against a commonly accepted and effective approach.

**Historical Negative Sampling (Historical)** [26] is chosen because it has proven effective in capturing the temporal dynamics of node pairs that have interacted before. Its ability to leverage past interactions makes it a strong reference point for evaluating models, especially in dynamic environments where repeated interactions carry important signals.

We attempted to include curriculum learning based negative sampling strategies [6, 12], but their limited reproducibility based on the provided resources prevented incorporation in our experiments.

*6.2.2 Dataset split.* To avoid potential bias introduced by dataset splitting and to account for the varying interaction frequencies in meme stock-related social networks over different periods, we employed time series cross-validation. The implementation involved dividing the dataset by month, where each month's data was used for training, and the following month's data was used for validation and testing. For the GME dataset, due to the large volume of interactions in January, February, and March, we further subdivided these months. The specific method for dataset splitting and the data volume for each month after the split, can be found in Appendix A.

*6.2.3 Evaluation.* To evaluate the performance, we tested each model using seven different types of negative samples. The type Random Sender and Random Receiver are generated by negative sampling strategy random sender and receiver. The type Loop are generated using negative sampling strategy negative loop. Type 6h, 12h, and 24h are derived from temporal sampling, with $t_n$ fixed at 72, 144, and 288 timestamps, respectively, to assess the model's ability to predict relationships between previously interacting nodes over 6h, 12h, and 24h intervals. Type Overall includes all of the aforementioned negative samples. For each type, we used all positive samples in the test set along with the corresponding negative samples for the specific category.

The evaluation metric we selected is the AUC (Area Under the Receiver Operating Characteristic Curve)[17]. We chose AUC because it provides a robust measure of a model's ability to distinguish between positive and negative samples, regardless of class imbalance. AUC evaluates the trade-off between true positive and false positive rates, making it particularly suitable for our tasks where the ratio of positive to negative samples can vary significantly [21]. AUC is also used by most DGE studies [7, 29, 38, 42].

*6.2.4 Hyper-parameter Setting.* The $q$ and $t_f$ for temporal sampling in our proposed negative sampling strategy, DINS, is set to 5 and 288 timestamps (1 day) respectively for all experiments. The hyper-parameter settings of the three DGE models (see Appendix B) are based on the original paper and fine tuned on our datasets.

## 6.3 Effect of Negative Sampling Strategy

In this subsection, we analyse the effect of negative sampling strategies in both evaluating and training DGE models, respectively. We use three negative sampling strategies: random, historical, and DINS (our proposed negative sampling strategy ) to train three DGE models: TGNs, DyGFormer, and GraphMixer on three datasets, BB, AMC, and GME. This results in a total of 27 experiments.

*6.3.1 Evaluation.* We first show the effect of using different negative sampling strategies for evaluation on the same model (trained with the same negative sampling strategy on the same dataset). In Figure 2, we present the results of DyGFormer on the three datasets, showing how its performance varies with different negative sampling strategies used during evaluation. The results for the other two models can be found in the Appendix C. The experimental results are similar for all three DGE models.

DyGFormer trained with random negative sampling strategy (blue bars) achieves high AUC scores when evaluated with Random Receiver-type negative samples. However, when evaluated with 6h-type negative samples, the AUC falls below 0.5 in most cases, except for BB in January. Although AUC scores are a little bit higher when evaluated using other types of negative samples, the performance remains below that of Random receiver-type negative samples. For DyGFormer trained with historical negative sampling strategy (green bars), the model shows higher AUC scores when evaluated with 6h-type negative samples, but similar to the previous case, AUC significantly drops when evaluated with other types of negative samples. DyGFormer trained with our proposed negative sampling strategy DINS (orange bars) performs well across most evaluation types except Random receiver on some months. This highlights the need for evaluation with different and diverse sampling strategies, anchored in the domain knowledge.

Additionally, as we discussed in Section 5, each of these proposed negative sampling strategies plays an important role in evaluating different aspects of meme stock-related social networks. Since the model's performance varies depending on the type of negative samples used for evaluation, our proposed negative sampling strategies provides a more comprehensive evaluation framework. This allows us to capture a fuller range of interaction dynamics in meme stock-related social networks, ensuring that models are tested more thoroughly across multiple dimensions of the network's behavior.

*6.3.2 Training.* Next, we show the effect of negative sampling strategy used in training again starting with the results on DyGFormer. When evaluating with overall-type negative samples, the model trained using DINS consistently outperforms those trained with the two baseline strategies across all datasets. This suggests that incorporating our proposed strategy during training enhances the model's overall predictive ability for meme stock-related social networks. We then examine performance across various types of negative samples. When evaluated with random receiver-type negative samples, the model trained using DINS shows slightly

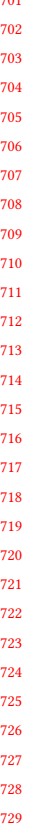
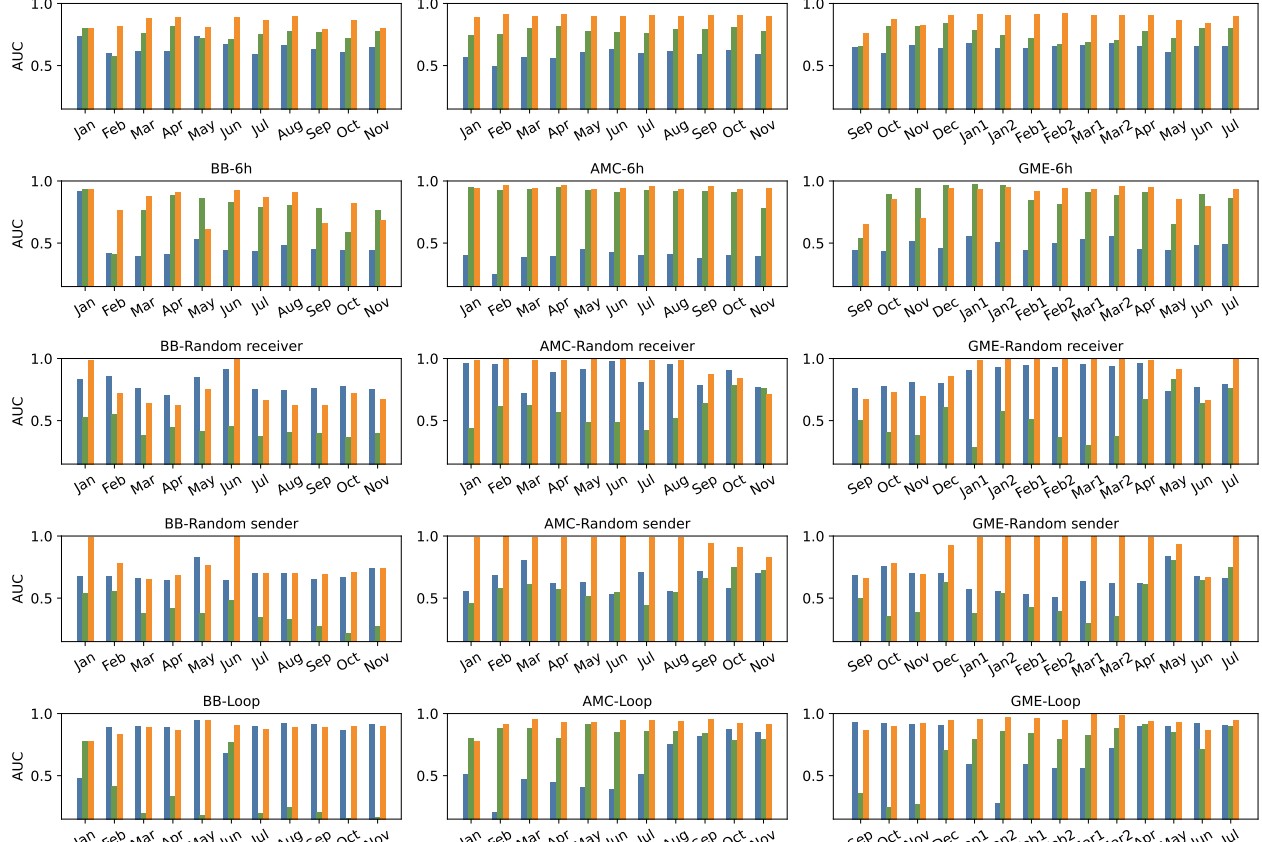

**Figure 2: The evaluation results of DyGFormer trained with three different negative sampling strategies. The blue bar, green bar and orange bar show the results of model trained with random negative sampling strategy, historical negative sampling strategy and DINS (proposed).**

**Table 3: Overall rank of DGE models trained using different negative sampling strategies across three datasets. Bold indicates the best rank.**

| Strategy | Dataset | TGNs | DyGFormer | GraphMixer |
|----------|---------|------|-----------|------------|
| Random | | 3.00 | 2.81 | 2.63 |
| Historical | BB | 2.00 | 2.09 | 2.09 |
| DINS | | **1.00** | **1.09** | **1.27** |
| Random | | 1..90 | 3.00 | 2.54 |
| Historical | AMC | 3.00 | 2.00 | 2.18 |
| DINS | | **1.09** | **1.00** | **1.27** |
| Random | | 2.28 | 3.00 | 2.50 |
| Historical | GME | 2.57 | 2.00 | 2.07 |
| DINS | | **1.14** | **1.00** | **1.42** |

lower AUC than the model trained with random negative sampling strategy. This is expected, as the random negative sampling strategy focuses primarily on distinguishing this specific type of negative sample. However, for other types of negative samples, the model trained with DINS consistently demonstrates significantly better performance compared to those trained with random

negative sampling. When comparing to the model trained with historical negative sampling, the model trained with DINS shows higher performance across all types of negative samples.

Following, we analyze the impact of negative sampling strategies on the training of all DGE models. Detailed results can be found in Table 3. The data in the table represents the average ranking of each model and negative sampling strategy combination across the monthly splits of each dataset. The results show that the proposed negative sampling strategy consistently outperforms the other two baselines across all DGE models and datasets. Notably, the proposed strategy demonstrates a significant advantage with DyGFormer and TGNs, almost always achieving the top rank. Considering that these three models use different designs, we believe that proposed negative sampling strategy can enhance the prediction ability of various DGE models on meme stock-related social networks.

## 6.4 Ablation Study

In this section, we conduct an ablation study using the BB dataset and DyGFormer to validate the impact of individual negative sampling strategies. We remove each individual strategy from DINS and

**Table 4: AUC of ablation studies for negative sampling strategies for DyGFormer on BB dataset.**

| Month | Jan | Feb | Mar | Apr | May | Jun | July | Aug | Sep | Oct | Nov |
|---|---|---|---|---|---|---|---|---|---|---|---|
| Proposed | 0.7990 | 0.8135 | 0.8779 | 0.8872 | 0.8073 | 0.8917 | 0.8620 | 0.9016 | 0.7897 | 0.8644 | 0.7971 |
| - temporal | 0.7721 | 0.8021 | 0.7952 | 0.7916 | 0.7921 | 0.7597 | 0.7813 | 0.7865 | 0.7646 | 0.7436 | 0.7617 |
| - loop | 0.8620 | 0.8093 | 0.7971 | 0.8856 | 0.8063 | 0.8209 | 0.8371 | 0.8307 | 0.8520 | 0.8359 | 0.7846 |
| - sender | 0.8968 | 0.9132 | 0.8276 | 0.8203 | 0.8552 | 0.7426 | 0.7262 | 0.7996 | 0.7492 | 0.6014 | 0.7322 |

**Table 5: Running time of one epoch for training different DGE models using different negative sampling strategies on the AMC-Jan dataset.**

| Strategy | TGNs | DyGFormer | GraphMixer |
|---|---|---|---|
| Random | 2264 s | 523 s | 112 s |
| Historical | 2297 s | 502 s | 113 s |
| Proposed | 2484 s | 1015 s | 271 s |

train the model with the remaining strategies. The performance is evaluated using overall-type negative samples. The results are shown in Table 4. The ablation study of other two DGE models is shown in Appendix D which show similar results.

Removing temporal sampling and negative loop sampling led to performance drops across all monthly datasets, with the only exception being a small improvement in January when negative loop sampling was removed, further confirming that both strategies are meaningful and necessary. For random sender, its removal leads to significant improvements in January, February, and May datasets, but results in notable declines for other months. Moreover, in January, February, and May, the model loses predictive power for the random sender negative sample type (AUC approximately 0.5). Given the importance of random sender samples in predicting meme stock-related social networks, we conclude that random sender sampling remains both meaningful and necessary.

### 6.5 Running Time

In this subsection, we discuss the additional training time caused by the proposed negative sampling strategy. Although the sampling process itself is not overly complex, the proposed strategy increases the number of negative samples. Therefore, our primary focus is on the training time rather than the complexity of the sampling process. The time needed for one epoch of training three different DGE models using three different negative sampling strategies on the AMC-Jan dataset is shown in Table 5.

For the random and historical sampling strategies, the training times for all models are relatively similar. However, when using DINS, there is an increase. For DyGFormer and GraphMixer, the time required roughly doubles, while for TGNs, the increase is less pronounced, adding only a small amount of additional time. Although the proposed strategy results in a longer training time, especially for DyGFormer and GraphMixer, this increase is compensated by the significant improvement in model performance.

### 6.6 Discussion

Through experiments, we demonstrated the critical impact of negative sampling strategies in training and evaluating DGE models

for predicting meme stock-related social networks. We validate that our proposed negative sampling strategies offers a more comprehensive and accurate evaluation compared to existing methods. Furthermore, we validated that training DGE models using our proposed DINS significantly enhances their ability to predict interactions in meme stock-related social networks. The results showed that, for SOTA DGE model DyGFormer trained using our proposed DINS, the AUC scores consistently reached high levels, with values of at least 0.8 across various datasets. In existing studies, AUC are typically above 0.9 because these models are trained and evaluated with random negative sampling which focus on a single strategy in contrast to our approach which consists of a variety of strategies.

This strong predictive performance indicates that our approach can contribute to a more effective analysis of meme stock-related social networks, which can, in turn, help in understanding meme stock price movements. Since these networks play a critical role in driving stock price volatility through online discussions and collective sentiment shifts, the ability to accurately model and predict user interactions within these networks provides valuable insights into potential market behavior. By improving the predictive power of DGE models, our proposed strategy could assist in identifying key patterns and interactions that may correlate with significant changes in meme stock prices.

### 7 Conclusion

In this paper, we study the use of DGE models in predicting user interactions on meme stock-related social networks. We analyzed three real-world meme stock-related Reddit networks and demonstrated that the current design of negative sampling strategies is insufficient for DGE models to accurately predict interactions. To address this issue, we proposed several individual domain knowledge-informed negative sampling strategies and presented a method, DINS, to combine these individual strategies effectively during training. The experiments showed that our proposed negative sampling strategies can better evaluate the ability of DGE models in analysing meme stock-related social networks and improve their predictive performance.

Our future work will explore the practical application of DGE models optimized for meme stock-related social network analysis, with the aim of gaining deeper insights into meme stock prices. In addition, since the proposed negative sampling strategy increases the training time, we will explore integration of active learning techniques to address this downside.

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

## A    Describable statistics of split datasets

The describable statistics of the monthly split subsets for the BB, AMC, and GME datasets are shown in Table 6. Each row represents the interactions occurring in the training set for a given month, while the validation and test sets contain all interactions that occurred in the following month. Due to the transductive setting, all interactions in the validation and test sets involving nodes that are not present in the training set are removed. The table shows the sizes of the validation and test sets after this adjustment. An exception is made for GME from January to March, where the specific time periods of the interactions included in each subset are detailed in the "Date" column.

From Table 6, it can be observed that the proportion of the training set varies across different subsets. This variation allows us to analyze how the size of the training set influences the model's performance on subsequent interactions.

**Table 6: Descriptive statistics of the monthly split subsets for the BB, AMC, and GME datasets.**

| Month | Training | Validation & Test | Total | Training Set Percent | Date |
|---|---|---|---|---|---|
| **BB** | | | | | |
| Jan | 127,634 | 22,041 | 149,675 | 85.27% | |
| Feb | 29,342 | 6,208 | 35,550 | 82,53% | |
| Mar | 8,479 | 7,033 | 15,512 | 54.66% | |
| Apr | 9,197 | 5,671 | 14,868 | 61.85% | |
| May | 9,857 | 64,755 | 74,612 | 13.21% | |
| Jun | 175,433 | 4,872 | 180,305 | 97.29% | |
| Jul | 5,321 | 3,726 | 9,047 | 58.81% | |
| Aug | 4,838 | 3,918 | 8,756 | 55.25% | |
| Sep | 5,280 | 4,104 | 9,384 | 56.26% | |
| Oct | 5,250 | 3,640 | 8,890 | 59.05% | |
| Nov | 4,654 | 3,540 | 8,194 | 56.79% | |
| **AMC** | | | | | |
| Jan | 218,408 | 272,444 | 490,852 | 44,50% | |
| Feb | 441,803 | 117,123 | 558,926 | 79,04% | |
| Mar | 138,658 | 108,147 | 246,805 | 56,18% | |
| Apr | 122,918 | 164,284 | 287,202 | 42,80% | |
| May | 215,970 | 276,249 | 492,219 | 43,88% | |
| Jun | 363,492 | 135,331 | 498,823 | 72,87% | |
| Jul | 143,735 | 110,627 | 254,362 | 56,51% | |
| Aug | 120,137 | 66,485 | 186,622 | 64,37% | |
| Sep | 73,111 | 43,967 | 117,078 | 62,45% | |
| Oct | 48,722 | 41,267 | 89,989 | 54,14% | |
| Nov | 45,805 | 38,739 | 84,544 | 54,18% | |
| **GME** | | | | | |
| Sep | 2,024 | 6,041 | 8,065 | 25,10% | |
| Oct | 12,113 | 6,721 | 18,834 | 64,31% | |
| Nov | 13,251 | 35,790 | 49,041 | 27,02% | |
| Dec | 50,057 | 131,143 | 181,200 | 27,63% | |
| Jan1 | 243,703 | 558,405 | 802,108 | 30,38% | 01.01.2021 - 24.01.2021 |
| Jan2 | 895,344 | 555,120 | 1,450,464 | 61,73% | 25.01.2021 - 29.01.2021 |
| Feb1 | 662,931 | 402,929 | 1,065,860 | 62,20% | 30.01.2021 - 14.02.2021 |
| Feb2 | 464,310 | 490,503 | 954,813 | 48,63% | 15.02.2021 - 28.02.2021 |
| Mar1 | 525,218 | 463,119 | 988,337 | 53,14% | 01.03.2021 - 15.03.2021 |
| Mar2 | 474,670 | 275,869 | 750,539 | 63,24% | 16.03.2021 - 31.03.2021 |
| Apr | 290,751 | 33,120 | 323,871 | 89,77% | |
| May | 52,559 | 42,196 | 94,755 | 55,47% | |
| Jun | 65,572 | 36,135 | 101,707 | 64,47% | |
| Jul | 23,839 | 16,830 | 40,669 | 58,62% | |

## B    Hyperparameter Setting

In this section, we present the hyperparameter settings of the three dynamic graph embedding methods. The hyperparameters are designed based on the original paper and fine tuned on our datasets. Please refer the original papers [7, 29, 42] and codes[1], for the specific meaning of the hyperparameters.

*Hyperparameter for TGNs.* The batch size is set at 1000, learning rate at 0.0001, memory dimension at 172, number of heads at 2, number of layers at 1, dropout rate at 0.1, number of neighbors at 10, embedding module used at graph attention, memory updater at GRU, aggregator at last, message function at identity, and embedding module at graph attention.

*Hyperparameter for DyGFormer.* The batch size is set at 1000, learning rate at 0.0001, channel embedding dimension at 50, patch size at 2, number of layers at 2, number of heads at 2, and dropout rate at 0.1.

*Hyperparameter for GraphMixer.* The batch size is set at 1000, learning rate at 0.0001, number of tokens at 20, number of layers at 2, and dropout rate at 0.1.

## C    Time series prediction results of TGNs and GraphMixer

In Figure 3 and Figure 4, we present the time series prediction results of GraphMixer and TGNs on the three datasets, showing how its performance varies with different negative sampling strategies used during training and evaluation.

## D    Ablation Study

In this section, we conduct an ablation study using the BB dataset and GraphMixer and TGNs to validate the impact of individual negative sampling strategies. We remove each individual strategy from DINS and train the model with the remaining strategies. The performance is evaluated using overall-type negative samples. The results are shown in Table 7.

---

[1]The TGNs is implemented based on codes provided at https://github.com/twitter-research/tgn and DyGFormer and GraphMixer is implemented based on code provided at https://github.com/yule-BUAA/DyGLib

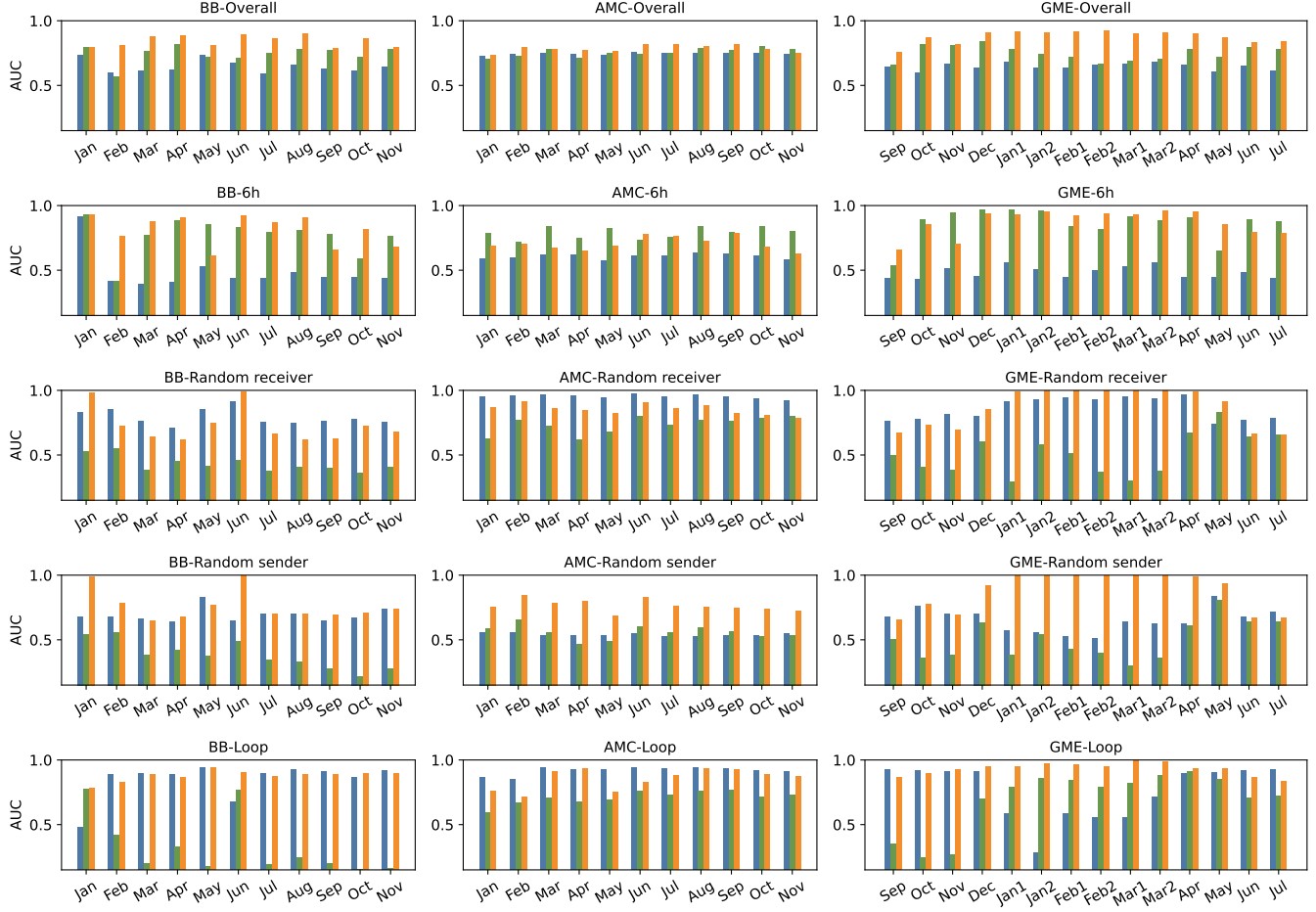

**Figure 3: The evaluation results of GraphMixer trained with three different negative sampling strategies. The blue bar, green bar and orange bar show the results of model trained with random negative sampling strategy, historical negative sampling strategy and DINS (proposed).**

**Table 7: AUC of ablation studies for negative sampling strategies for GraphMixer and TGNs on BB dataset.**

| Ablation-TGN | Jan | Feb | Mar | Apr | May | Jun | July | Aug | Sep | Oct | Nov |
|---|---|---|---|---|---|---|---|---|---|---|---|
| Proposed | 0.7892 | 0.8060 | 0.8022 | 0.7632 | 0.7949 | 0.8312 | 0.8216 | 0.7914 | 0.8091 | 0.8082 | 0.8218 |
| ablation - loop | 0.7592 | 0.7814 | 0.7850 | 0.7468 | 0.7903 | 0.8332 | 0.8124 | 0.7836 | 0.7984 | 0.8180 | 0.8165 |
| ablation - future | 0.7307 | 0.7374 | 0.6966 | 0.7051 | 0.7648 | 0.7660 | 0.7755 | 0.7164 | 0.7746 | 0.7245 | 0.7403 |
| ablation - sender | 0.7944 | 0.8336 | 0.8197 | 0.8074 | 0.8088 | 0.8116 | 0.8292 | 0.8219 | 0.8228 | 0.8236 | 0.8417 |
| Ablation-GraphMixer | Jan | Feb | Mar | Apr | May | Jun | July | Aug | Sep | Oct | Nov |
| Proposed | 0.7096 | 0.7485 | 0.7579 | 0.7683 | 0.7574 | 0.7554 | 0.7721 | 0.7855 | 0.7803 | 0.7435 | 0.7864 |
| ablation - loop | 0.6929 | 0.7503 | 0.7349 | 0.7619 | 0.7197 | 0.7424 | 0.7558 | 0.7788 | 0.7675 | 0.7604 | 0.7794 |
| ablation - future | 0.6930 | 0.6576 | 0.7211 | 0.6694 | 0.7112 | 0.7023 | 0.6136 | 0.7000 | 0.5427 | 0.6747 | 0.6734 |
| ablation - sender | 0.7602 | 0.8271 | 0.8030 | 0.7780 | 0.7752 | 0.7882 | 0.7792 | 0.6875 | 0.7873 | 0.7722 | 0.6149 |

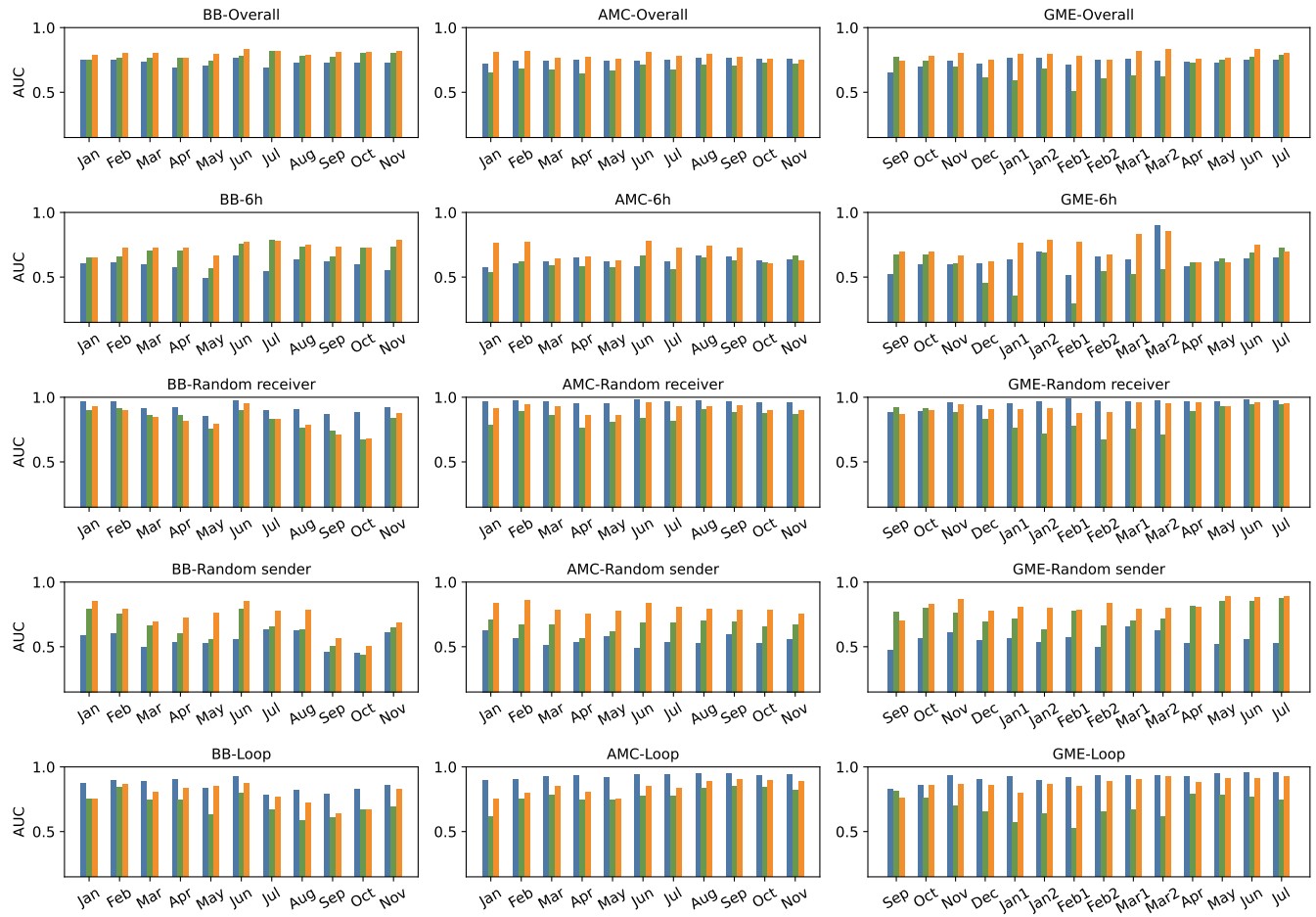

Figure 4: The evaluation results of TGNs trained with three different negative sampling strategies. The blue bar, green bar and orange bar show the results of model trained with random negative sampling strategy, historical negative sampling strategy and DINS (proposed).

