# OpenReview forum: "Domain-Informed Negative Sampling Strategies for Dynamic Graph Embedding in Meme Stock-Related Social Networks"
_ACM.org/TheWebConf/2025/Conference — WWW 2025 Oral_

### Official Review · Reviewer_MeeD · 2024-11-08

**Novelty:** 6
**Technical Quality:** 5

**Review:**

This paper proposes a set of domain-informed negative sampling strategies (DINS) for training dynamic graph embedding (DGE) models, focusing on the unique challenges posed by meme stock-related social networks, particularly those formed on platforms like Reddit. The paper introduces multiple individual negative sampling strategies such as temporal sampling, random sender/receiver, and negative loops, and combines them into the DINS strategy to improve the performance of models like TGNs, DyGFormer, and GraphMixer. The authors claim that DINS can better evaluate and train models to predict interactions in dynamic meme stock-related social networks.

Authors provide Comprehensive Evaluation for their approach. The paper evaluates the proposed strategy using a variety of baselines (e.g., random negative sampling, historical sampling) across three datasets (GME, AMC, BB), and it shows that the DINS strategy improves model performance in predicting future interactions in meme stock-related social networks.

## Weaknesses and Areas for Improvement:

1.	**Lack of Experiment Replication:** One concern is that the paper does not clearly mention how many times the experiments were repeated. Given the potential for random variance in graph-based models and sampling strategies, more details are needed on whether the experiments were run multiple times to validate the stability of the results. Were the results statistically significant, or could they be due to random fluctuations?
2.	**Specificity of the Domain:** The paper could benefit from a clearer explanation of the domain-specific characteristics of meme stock-related social networks. While the paper mentions the importance of predicting future interactions and their impact on stock prices, it is not fully clear what makes these networks distinctly different from other social networks in the context of DGE models. Could the authors clarify the specific aspects of meme stock-related social networks that make their negative sampling strategy particularly effective?

## Conclusion
The paper presents a strong and innovative approach to improving dynamic graph embedding models by incorporating domain-specific knowledge in the form of negative sampling strategies. The methodology has shown potential to enhance prediction accuracy in meme stock-related social networks, which could have significant implications for understanding and predicting stock market behaviors. However, clarification is needed regarding the experiment replication and statistical significance of the results, as well as the precise characteristics of meme stock-related social networks that make the proposed methods particularly effective. With further clarification and additional details, the contributions of this paper could be strengthened.

**Questions:**

1) Experiment Replication and Statistical Significance:
*	How many times were the experiments in Table 4 repeated? Can you clarify whether the results reported are consistent across multiple runs, and if so, what the standard deviation or confidence intervals are for the reported AUC scores?
*	Were the results statistically significant, and how did you ensure that the performance improvements observed were not due to random variations in the data?

2) Specificity of the Domain:
*	Could you elaborate on the domain-specific features of meme stock-related social networks that make your negative sampling strategy (DINS) particularly effective for this context?
*	How do these features differ from typical social networks, and how do they influence the interactions between users in these networks?
*	Can you provide more concrete examples of how the dynamics in meme stock-related social networks differ from other dynamic graph datasets, and how your approach takes these differences into account?

3) Comparison with stronger baselines:	You compare DINS with random and historical negative sampling strategies. Have you considered comparing DINS with other state-of-the-art negative sampling methods (e.g., adversarial or model-aware approaches)?

4) Handling Rare Events or Outliers: In meme stock-related social networks, some interactions or nodes may be rare or outliers, potentially leading to noise. How does your negative sampling strategy handle rare events or outliers in the data? Does DINS offer any advantages in dealing with these challenges compared to traditional methods?

**Reviewer Confidence:**

3: The reviewer is confident but not certain that the evaluation is correct

**Scope:**

4: The work is relevant to the Web and to the track, and is of broad interest to the community

---

### Official Review · Reviewer_T9tF · 2024-11-13

**Novelty:** 5
**Technical Quality:** 3

**Review:**

This paper addresses a dynamic link prediction (DLP) problem in meme stock-related social networks. Authors use a dynamic graph embedding (DGE) approach to predict links and show how to improve negative sampling technique. They show why current strategies works bad - mainly it's sampling edges at random - and suggest some more appropriate tools to capture repeptitive connections of users in a network. To prove the success of their strategies, they collect 3 datasets from Reddit connected to meme stocks and experiment with 3 modern DGE algorithms by adding the developed sampling strategies. Results show that AUC for link prediction increases.

**Questions:**

* Authors should clarify whether the experimental results are statistically significant. Did they run it multiple times and how big was the diversity?
* Are the results specific to the considered type of networks, or the proposed method (DINS) can improve DLP in other domains?
* It remains unclear how the improvement of DLP helps to predict stock price movements.
* Is there an explanation why (in Table 4) the removal of random sender gives such significant AUC improvements in January, February (from 80 to 90%)?

**Reviewer Confidence:**

3: The reviewer is confident but not certain that the evaluation is correct

**Scope:**

4: The work is relevant to the Web and to the track, and is of broad interest to the community

---

### Official Review · Reviewer_W7s4 · 2024-12-01

**Novelty:** 5
**Technical Quality:** 6

**Review:**

This paper presented a novel joint negative sample strategies for dynamic graph embedding. This joint strategy consists of randomizing sender and receiver, temporal negative sampling, negative loop sampling, and a positive sample enhance to balance the ratio between positive and negative samples, overcoming the problem of DGS in accurately predicting the re-interacted nodes. Experimental results show that the proposed method is very effective in terms of performance and time efficiency against the selected baselines.

Strength:

1. The proposed negative sampling method is novel and effective.

2. The experiments are solid, and the results are convincing.

3. The paper is well-organized, which is easy to understand.

Weakness:

1. Some claims lack evidence. For example, the authors claim that the performance of DGE for practical uses in real applications such as meme stock-related social network prediction is low but no evidence is provided.

2. The design of randomizing sender and receiver, temporal negative sampling, and negative loop sampling are used to address the highly dynamics of meme stock-related social networks. Additionally, the authors also point out that existing negative sampling strategies are designed for conventional dynamic link prediction, which fails to capture the specific patterns present in meme stock-related social networks. I am wondering what specific patterns present in meme stock-related social networks are identified and how the proposed joint negative sampling method addresses them. I believe an extra section to analyze and identify the specific patterns present in meme stock-related social networks would be helpful.

3. The authors argue that existing DGEs [11, 29, 45] do not work well in meme stock-related social network prediction while only benchmarks against [29]. [11, 45] should serve as baselines to benchmark the proposed method.

4. One main argument the authors made is that the DGE model is easy to be dominated by extremely large amounts of negative samples and leads to poor performance when learning from the positive samples. As a result, the authors proposed a positive sampling to balance the ratio between positive and negative samples, overcoming the data imbalance issues, which is great. However, the effectiveness of the positive sampling has not yet been reported. I suggest testing every negative sampling strategy with and without the positive sampling strategy in the ablation studies such that we can know how well the positive sampling strategy works.

**Questions:**

1. What are the specific and unique patterns present in meme stock-related social networks?

2. How does each of the proposed negative sampling strategy respectively address the unique patterns present in meme stock-related social networks?

3. How effective is the proposed positive sampling strategy in solving the data imbalance issues?

**Reviewer Confidence:**

4: The reviewer is certain that the evaluation is correct and very familiar with the relevant literature

**Scope:**

4: The work is relevant to the Web and to the track, and is of broad interest to the community

---

### Official Review · Reviewer_q3ph · 2024-12-03

**Novelty:** 4
**Technical Quality:** 4

**Review:**

The paper investigates how to better analyze and predict interactions in meme stock-related social networks, such as Reddit's WallStreetBets. It focuses on improving Dynamic Graph Embedding (DGE) models by proposing new negative sampling strategies tailored to such networks.

Strengths:
This paper introduces domain-specific insights into a technical framework, addressing a critical gap in current DGE methods.
This paper introduces a composite negative sampling strategy named DINS that combines these individual strategies and balances positive and negative samples through positive enhancement.
Weaknesses:
The proposed strategies require more computational resources, as evidenced by higher training times compared to baseline approaches. While effective, the additional overhead may limit scalability for extremely large datasets.
The ablation study is limited in quantitatively explaining the relative importance of each individual negative sampling strategy.
The authors have not provided open-source code.

**Questions:**

Ablation experiments indicate that the performance improvements vary significantly across different ablations. Could the authors provide a deeper analysis of why this phenomenon occurs?

**Reviewer Confidence:**

2: The reviewer is willing to defend the evaluation, but it is likely that the reviewer did not understand parts of the paper

**Scope:**

3: The work is somewhat relevant to the Web and to the track, and is of narrow interest to a sub-community

---

### Official Review · Reviewer_AoU9 · 2024-12-04

**Novelty:** 5
**Technical Quality:** 5

**Review:**

This paper introduces domain-informed negative sampling strategies for Dynamic Graph Embedding (DGE) models, specifically tailored for meme stock-related social networks. It addresses the limitations of conventional negative sampling by proposing novel strategies that consider the unique dynamics of these networks, such as interaction timing and types. The proposed methods aim to enhance the predictive performance of DGE models, which is crucial for understanding market trends and stock price movements. The paper demonstrates through experiments that these strategies significantly outperform existing baselines, offering a more accurate evaluation and training framework for DGE models in the context of meme stocks.

Strengths:

S1. The paper introduces an innovative approach by integrating domain-specific knowledge into negative sampling strategies, which is significant for enhancing the performance of Dynamic Graph Embedding (DGE) models in specific social network environments.

S2. The authors propose a negative sampling strategy (DINS) that effectively combines various sampling methods. This comprehensive approach allows for a more holistic assessment of DGE models, particularly in predicting complex interactions within social networks.

S3. The paper's proposed strategies have been experimentally validated to significantly enhance model predictive performance, offering an effective tool over existing methods in social network analysis and financial forecasting.

Weaknesses:

W1. The negative sampling strategies proposed in the paper perform exceptionally well on meme stock-related social networks, but readers may be interested in the applicability of these strategies to other types of social networks or different financial contexts. It is recommended that the authors further discuss the generalization potential of these strategies in the discussion section, possibly supporting their views with comparative analysis or theoretical arguments. Additionally, the authors might consider conducting additional experiments on other datasets to demonstrate the model's generalization capabilities.

W2. Given that the proposed negative sampling strategies may increase the computational burden of model training, it is suggested that the authors provide more detailed information on training time and resource consumption (such as memory and CPU usage) in the experimental section.

**Questions:**

Please see W1 & W2.

**Reviewer Confidence:**

3: The reviewer is confident but not certain that the evaluation is correct

**Scope:**

4: The work is relevant to the Web and to the track, and is of broad interest to the community